# High-Throughput DNA Metabarcoding as an Approach for Ichthyoplankton Survey in Oujiang River Estuary, China

**Rijin Jiang [1], James Leonard Lusana [2,3,]\*** and **Yongjiu Chen [2,]\***

1    Zhejiang Marine Fisheries Research Institute, Zhoushan 316021, China
2    College of Marine Science and Technology, Zhejiang Ocean University, Zhoushan 316022, China
3    School of Aquatic Sciences and Fisheries Technology, University of Dar es Salaam,
     Dar es Salaam P.O. Box 60091, Tanzania
\*    Correspondence: lusanaj@gmail.com or lusana.james@udsm.ac.tz (J.L.L.); yongjiuchen@zjou.edu.cn or
     yongjiu.chen@gmail.com (Y.C.)

**Abstract:** High-throughput DNA metabarcoding of mitochondrial 12S rRNA and Cyt b gene sequences was coupled with a morphology-based identification tool to assess ichthyoplankton community structure in Oujiang River Estuary, China. The performances of 12S and Cyt b barcoding markers were compared in terms of taxonomic resolution, detection and coverage, and their suitability was established for use as a quick and powerful ichthyoplankton assessment tool. A total of 30,138 ichthyoplankton (2462 eggs and 27,676 larvae) samples were collected from April to August 2015 and identified to 145 taxa belonging to 57 families and 105 genera. June and July were the main spawning months. Ichthyoplankton were more abundant around Lingkun and Qidu Islands and the upper parts of Oujiang River Estuary. The 12S gene marker presented higher species coverage and detection rate than Cyt b. DNA metabarcoding exhibited more representative species identification power than morphology. The findings reported in this study provided a key attempt towards the development of time-efficient and cost-effective ichthyoplankton identification and assessment tool.

**Keywords:** ichthyoplankton; Oujiang River Estuary; metabarcoding; morphology; 12S; Cyt b

## 1. Introduction

Ichthyoplankton are the early life stages (eggs and larvae) of marine fishes found in the sunlight zone of the water column usually less than 200 m deep [1,2]. Ichthyoplankton research is important because it provides information about both juvenile and adult fishes, such as spawning seasons and locations, recruitment strength, migration history, and spatial and temporal structures [3,4]. This information is essential for effective fish stock management and conservation particularly in light of anthropogenic disturbances and rapid climate changes [2]. Correct and accurate identification of fish eggs and larvae is a crucial step for fish ecological studies and conservation planning. Misinterpretation of fish biology and ecology derived from inaccurate ichthyoplankton identification could lead to biased fish stock evaluations and subsequently, poor conservation and management policies [5,6].

The rapid advance of next-generation DNA sequencing (NGS) analysis [4,7,8] has revolutionized genetic approaches for biodiversity research by providing an alternative tool for fish identification and assessment across all life stages. NGS is often more cost-effective, rapid and accurate than traditional methods [7,9–12]. This technology is rapidly transforming aquatic research to the genomic level, and combatting various challenges in the marine environment, from food security and biodiversity loss to climate change [13].

DNA metabarcoding using NGS has recently emerged as a potentially powerful method for assessing and monitoring the community structure of fishes, including eggs and larvae [4,14]. In order to achieve higher and more accurate species resolution and detection, DNA metabarcoding requires a heedful selection of barcode markers and primers.

Conserved barcode markers targeting certain regions of the mitochondrial genes, e.g., 12S rRNA [11,15–17] and Cyt b [18] provide a broad taxonomic resolution and coverage for fishes, even when DNA is degraded or present at a very low concentration [19]. In particular, 12S has been recommended for animal metabarcoding due to the presence of highly conserved regions that flank variable regions, permitting the design of primers with high taxon resolution power for the aimed taxonomic groups, and allowing concurrent identification of massive sets of existing organisms in a single sample of pooled DNA [12,20].

The purpose of this study was to assess the spatial and temporal distribution, species identity and composition of ichthyoplankton community structure in Oujiang River Estuary using 12S and Cyt b DNA metabarcoding coupled with a morphology-based identification approach. The performances of 12S and Cyt b metabarcoding markers were also compared in terms of taxonomic resolution, detection and coverage, and their suitability was established for use as a quick and powerful ichthyoplankton assessment tool.

## 2. Materials and Methods

### 2.1. River Estuary Survey and Sample Collection

A total of five surveys were carried out across Oujiang River Estuary from April to August, 2015 using a commercial fishing boat. One survey was conducted every month for the period of four consecutive days during monthly tide flood. Fish eggs and larvae were collected using the shallow horizontal planktonic net or oblique drag sampling survey with a digital flow meter with the density index (Ind./100 m$^3$). Eleven sampling locations (F1–11) were established for ichthyoplankton surveys. Plankton samples were sieved through smaller sized meshes and washed with sea water, from which fish eggs and larvae were picked out, placed into separate jars according to sampling sites and months, and preserved in 100% ethanol prior to morphological and molecular analyses.

### 2.2. Morphological Assessment

Ichthyoplankton morphological identification followed [21] and used dissecting microscope attached with a camera (Nikon SMZ800- Tokyo, Japan). All fish eggs and larvae from all sampling sites and months were identified to the lowest possible taxonomic rank using morphological features.

### 2.3. Metabarcoding Assessment

2.3.1. DNA Extraction

A total of 22 tubes (samples) containing various eggs and larvae collected from six stations (F2, F3, F4, F5, F7, and F9) in all sampling months were sequenced for 12S and Cyt b DNA metabarcoding analysis. Total DNA was extracted using DNeasy® Blood and Tissue Kit (Qiagen, Valencia, CA, USA) from fish eggs and larvae in a batch of 15 samples. The resulting DNA samples were then pooled together for each specific site and month. Extracted DNA was visualized by agarose gel electrophoresis using 1% agarose gel in 1X TAE buffer stained with DNA Green fluorescent dye for band characterization through Gel Imaging System.

2.3.2. PCR Amplification and DNA Sequencing

A 12S gene fragment (<100 bp) was amplified by PCR using the primer set of teleo_F_L1848 and teleo_R_H1913 [12]. A second marker of Cyt b gene fragment (<460 bp) was amplified using the primer pair of L14841 and H15149 [18]. All PCRs were conducted in a Thermo Cycler with a 25 µL reaction volume containing 8.5 µL sterile nuclease-free water in analytical grade, 12.5 µL GoTaq® Green Master Mix (Promega Inc., Madison, WI, USA), 1 µL each of the primer set, and 2 µL template DNA. The thermal profile included a preliminary denaturation for 2 min at 95 °C followed by 35 cycles of denaturation at 95 °C for 30 s, annealing at 50 °C for 30 s, extension at 72 °C for 60 s and finally a single extra extension at 72 °C for 10 min. PCR products were confirmed via gel electrophoresis using

1.5% agarose gel in 1X TAE buffer stained with DNA Green fluorescent dye for band characterization through Gel Imaging System. PCR products for all genes were sent for NGS analysis at LC Science (Hangzhou, China) following standard protocols for PE300 library construction and sequencing on an Illumina MiSeq platform.

### 2.4. Data Analysis

### 2.4.1. Morphological Data

The spatial and temporal distribution patterns of eggs and larvae were visualized in 2D graphs produced in Surfer®. For quantitative analysis, the abundance of eggs and larvae was estimated by density, D (using D = N/V) in the number of individuals per cubic meter where N is the number of eggs/larvae per catch, and V is the filtration volume. Species percentage was used to measure the level of species dominance. The species with the highest percentage of total catch was considered the dominant species in each sampling period and area.

### 2.4.2. DNA Metabarcoding Data

All raw data from the MiSeq sequencing platform were received in FASTQ format and preprocessed by trimming the barcodes and the adapter sequences. Extended reads were produced by merging the paired ends of the sequences using FLASH software [22]. Trimmomatic [23] was performed on the merged data for quality filtering by discarding all tags that have an "N" base percentage higher than 5%, a low-quality base percentage $\geq$ 20% or a short sequence length. A quality control check was performed by visually analyzing a QC report generated in FastQC [24]. The resulting reads were then imported into the QIIME pipeline [25] using MacQIIME version 1.9.1 (http://www.wernerlab.org/software/macqiime (accessed on 26 October 2017). Low-quality reads and short sequences were removed; then clean reads were assigned to samples, or demultiplexed, based on their nucleotide barcode using the split_libraries_fastq.py script. The demultiplexed sequences were clustered into OTUs with CD-HIT at $\geq$97% similarities using the pick_otus.py script.

For both markers, a representative set of sequences were selected from each OTU using the pick_rep_set.py script. The 12S and Cyt b QIIME compatible databases were created in MacQIIME following the standard method by Baker [26] after downloading all available fish 12S and Cyt b sequences in the GenBank database. The representative sequences of each gene marker were then blasted against the created QIIME compatible reference database and assigned taxonomic names using the assign_taxonomy.py script (minimum percent identity = 95%, maximum e-value = 0.001). Finally, OTU tables were built (make_otu_table.py) and singletons were removed from OTUs (filter_otus_from_otu_table.py). The summarize_taxa_through_plots.py script was used to summarize species taxonomy.

A phylogenetic tree was constructed after aligning and filtering the representative set of sequences in MacQIIME. The community structure of fish eggs and larvae was determined by calculating within-community diversity (alpha diversity) and between-community diversity (beta diversity). The level of alpha diversity was determined by calculating Shannon (also known as Shannon-Weiner), Simpson and Chao1 indices (alpha_diversity.py). Beta diversity among sampling sites and months was compared for each metabarcoding gene using Bray-Curtis distance and visualized using principal coordinate analysis plots generated by beta_diversity_through_plots.py and make_2d_plots.py scripts.

## 3. Results

### 3.1. Morphology

Species Identification and Composition

A total of 30,138 (2462 eggs and 27,676 larvae) ichthyoplankton samples were collected and identified. The highest number of eggs and larvae were recorded in June (Table 1). Ichthyoplankton samples were morphologically classified into 38 fish groups, including eight groups of eggs and 30 groups of larvae. Twenty-eight groups were identified to species level, four groups to genus level and six groups to family level. *Coilia mystus*

(79.73%), *C. nasus* (11.86%), and Cyprinidae (7.55%) were the dominant fish egg species, and *C. nasus* (57.67%), *C. mystus* (33.30%), and Gobiidae (7.27%) were the dominant larva species in Oujiang River Estuary (Figure 1).

**Table 1.** Species number, quantity and average density of fish eggs and larvae in Oujiang River Estuary from April to August 2015.

| | Fish Eggs | | | Fish Larvae | | |
|---|---|---|---|---|---|---|
| | Number of Species | Number of Eggs | Average Density Eggs/100 m³ | Number of Species | Number Larvae | Average Density Larvae/100 m³ |
| April | 0 | 0 | 0 | 9 | 71 | 1.82 |
| May | 7 | 184 | 5.27 | 15 | 1997 | 46.65 |
| June | 4 | 1585 | 24.33 | 13 | 18,882 | 299.89 |
| July | 3 | 332 | 4.26 | 17 | 5054 | 66 |
| August | 3 | 361 | 5.19 | 12 | 1672 | 21.32 |
| Total | 9 | 2462 | 7.84 | 36 | 27,676 | 87.1 |

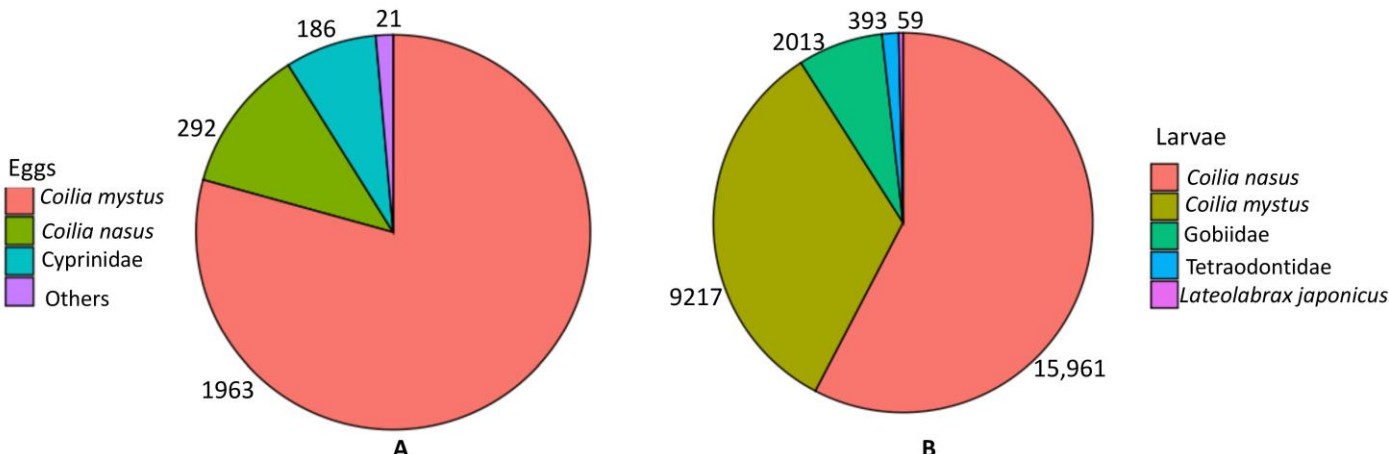

**Figure 1.** Number of dominant fish groups in Oujiang River Estuary. (**A**) Fish eggs and (**B**) fish larvae.

*3.2. Ichthyoplankton Spatial and Temporal Distribution*

Ichthyoplankton density varied among sampling points and seasons. In April, the average density of fish larvae was 1.82 ind./100 m³. Sampling station F8 (4.73 ind./100 m³), F7 (2.33 ind./100 m³) and F3 (2.27 ind./100 m³) had the highest fish larva densities. Other sampling stations had densities less than 2 ind./100 m³ (Figure 2). April was dominated by Mugilidae 3.59 ind./100 m³, *Lateolabrax japonicus* 3.56 ind./100 m³ and Engraulidae 2.63 ind./100 m³. No eggs were collected in April (Figure 3).

In May, the average density of fish larvae was 46.65 ind./100 m³. Sampling stations in the upper parts of Qidu Island, i.e., F10 (145.56 ind./100 m³), F1 (124.22 ind./100 m³) and F2 (110.78 ind./100 m³) had the greatest larva densities. Other sampling stations had larva densities less than 60 ind./100 m³. The dominant fish larvae in May were Gobiidae (254.24 ind./100 m³), *C. mystus* (90.22 ind./100 m³) and *C. nasus* (53.76 ind./100 m³) (Figure 2). The average egg density was 5.27 ind./100 m³ dominated by *C. mystus* (24.71 ind./100 m³), *C. nasus* (16.31 ind./100 m³) and *Cyprinus carpio* (8.05 ind./100 m³). The highest egg density was observed in the upper part of the river at sampling points of F1 (19.94 ind./100 m³), F4 (17.35 ind./100 m³) and F3 (14.12 ind./100 m³). Other sampling locations had egg densities less than 1 ind./100 m³ (Figure 3).

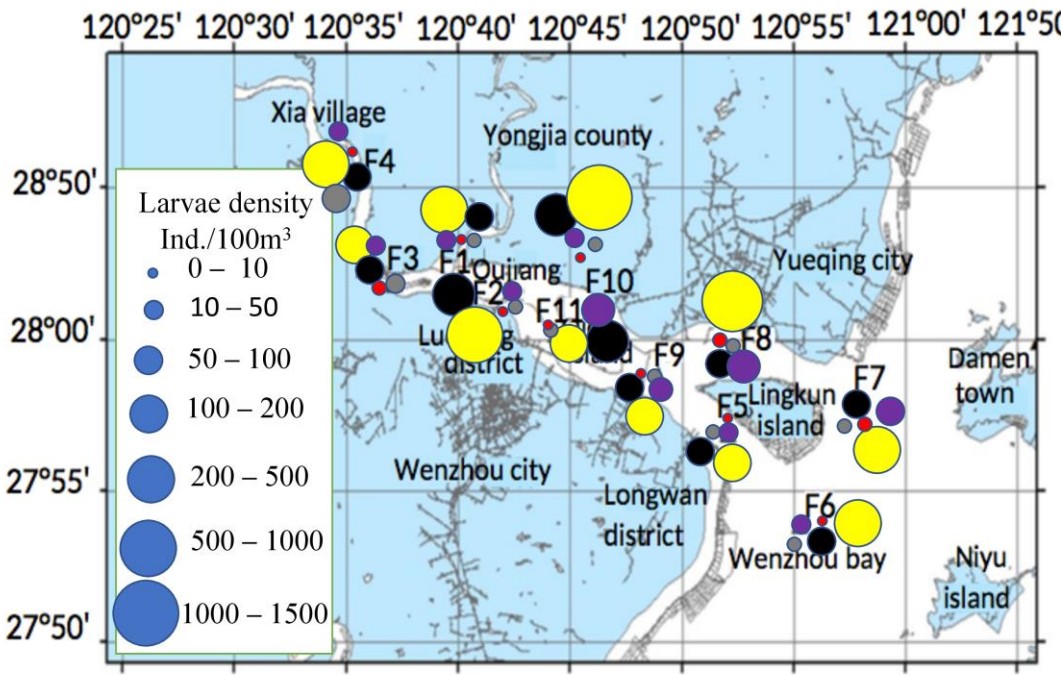

**Figure 2.** Spatial-temporal distribution of fish larvae in Oujiang River Estuary: each sampling month is indicated with a specific color: red = April, black = May, yellow = June, grey = July and purple = August. The size of circle reflects the density of larvae in a specific site.

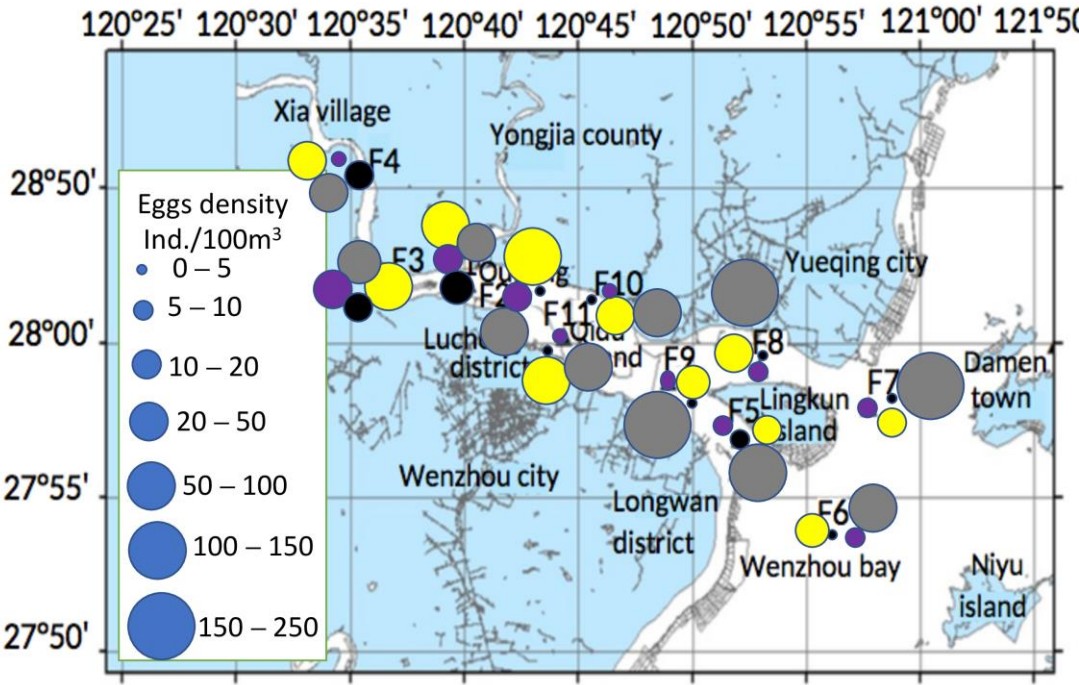

**Figure 3.** Spatial-temporal distribution of fish eggs in Oujiang River Estuary: Each sampling month is indicated with a specific color: black = May, yellow = June, grey = July and purple = August. The size of circle reflects the density of eggs in a specific site.

In June, the average density of 299.89 ind./100 m³ for fish larvae was recorded, dominated by *C. nasus* (2241.27 ind./100 m³) and *C. mystus* (743.28 ind./100 m³). The highest larva density was observed in areas around Qidu Island, i.e., F10 (1008.79 ind./100 m³), F11 (822.79 ind./100 m³) and F2 (755.18 ind./100 m³). Other sampling locations had larva densities less than 160 ind./100 m³ (Figure 2). The average egg density was 24.33 ind./100 m³

dominated by *C. nasus* (203.14 ind./100 m$^3$), Cyprinidae (25.61 ind./100 m$^3$) and *C. mystus* (13.76 ind./100 m$^3$). The highest egg density was observed in the upper part of the River at F2 (110.60 ind./100 m$^3$), F3 (53.16 ind./100 m$^3$) and F1 (48.13 ind./100 m$^3$). Other sampling locations had egg densities less than 30 ind./100 m$^3$ (Figure 3).

In July, the average density of 66 ind./100 m$^3$ for fish larvae was collected. Areas around Lingkun and Qidu Islands at F4 (37.01 ind./100 m$^3$) had the greatest larva densities. Other sampling stations had larva densities less than 30 ind./100 m$^3$. The dominant larvae were *C. mystus* (457.32 ind./100 m$^3$) and *C. nasus* (162.55 ind./100 m$^3$; Figure 2). The average egg density was 4.26 ind./100 m$^3$ dominated by *C. nasus* (46.7 ind./100 m$^3$) and *Stolephorus chinensis* (0.21 ind./100 m$^3$). The highest egg density was observed around the Lingkun Island at F8 (226.19 ind./100 m$^3$), F7 (175.88 ind./100 m$^3$) and F9 (164.24 ind./100 m$^3$; Figure 3).

In August, the average density of 21.32 ind./100 m$^3$ for fish larvae was collected. Sampling stations around Qidu Island to Lingkun Island, i.e., F8 (58.93 ind./100 m$^3$), F11 (44.54 ind./100 m$^3$), F7 (37.20 ind./100 m$^3$) and F9 (32.64 ind./100 m$^3$) had the greatest larva densities. The densities of other sites were less than 20 ind./100 m$^3$. The dominant fish larvae were *C. mystus* 172.90 ind./100 m$^3$ and *C. nasus* 42.62 ind./100 m$^3$ (Figure 2). The average eggs density was 5.9 ind./100 m$^3$ dominated by *C. mystus* 29.25 ind./100 m$^3$ and *C. nasus* 27.82 ind./100 m$^3$. Higher egg density stations were in upper parts of the River Estuary at F3 (24.12 ind./100 m$^3$), F2 (17.01 ind./100 m$^3$ and F1 (12.25 ind./100 m$^3$) (Figure 3).

*3.3. DNA Metabarcoding*

3.3.1. Sequencing and Reads Quality

The raw data generated libraries for 12S and Cyt b. Of 12S, 788,906 tags resulted from 79.81 Mb sequences. The quality control yielded 784,064 clean tags in 78.01 Mb, with an average GC content of 44.83% and the sequence length distribution of <200 bases. A second library of Cyt b generated 487,697 tags in 293.37 Mb. The quality control subsequently produced 433,420 tags in 148.48 Mb with an average 43.15% GC content that subsequently contributed to <315 bp sequence length distribution.

3.3.2. Species Identification and Composition

The DNA metabarcoding provided results about species identity, diversity, abundance, distribution, and composition of ichthyoplankton in Oujiang River Estuary. After taxonomic assignment of OTUs, about 0.01% of 12S sequences and 61.8% of Cyt b sequences had no BLAST hits. The 12S sequence dataset was assigned to 82 taxon groups from 661 OTUs with taxonomic coverage of 23 orders, 33 families, and 68 genera, of which 77 were identified to species level and five groups to genus level. The Cyt b metabarcoding recovered 412 OTUs, of which 46 taxa were identified to 22 orders, 34 families, and 51 genera. Within the 46 taxa, 45 were identified to species level and one taxon to genus level. The 12S metabarcoding analysis indicated that *C. nanus* (32.4%) and *C. mystus* (14.6%) were the dominant fish species, while the Cyt b metabarcoding revealed that *Moringua microchir* (20.6%) and *C. nanus* (10.7%) were the dominant fish species in Oujiang River Estuary, although 61% of the sequences had no BLAST hits and were not identified (Figure 4).

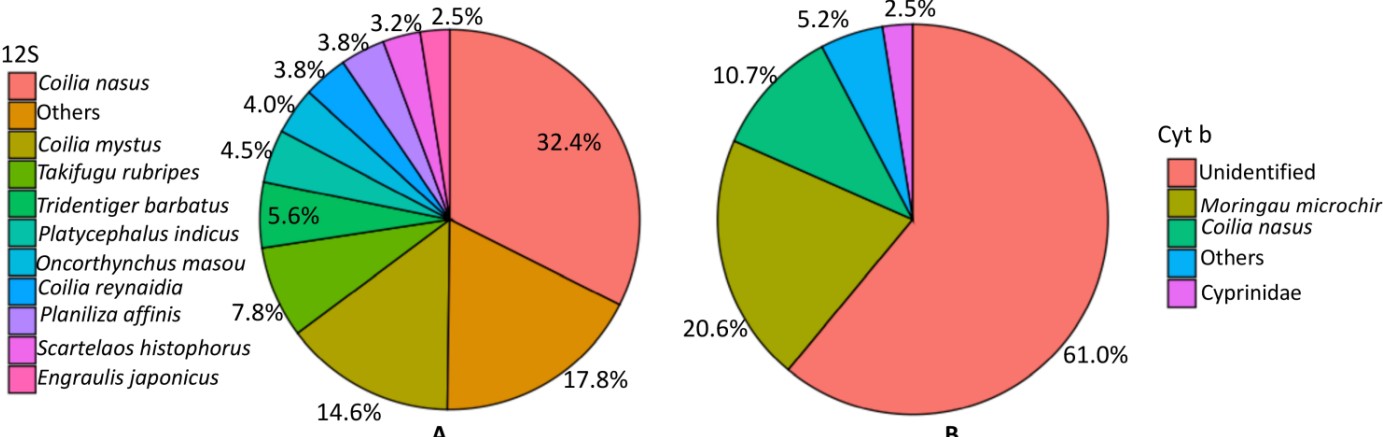

**Figure 4.** The species percentage composition of fish eggs and larvae in Oujiang River Estuary identified by molecular analysis: (**A**) 12S DNA metabarcoding, (**B**) Cyt b DNA metabarcoding.

### 3.3.3. Ichthyoplankton Community Structure and Diversity Patterns

In Oujiang River Estuary, the analysis of 12S alpha diversity reflected by Shannon and Simpson indices indicated that the highest species abundance and diversity were found in August, followed by July, June and April. F5 (July) located around Lingkun Island had the most diverse composition, followed by F5 (April), F3 (August), F9 (August) and F9 (June), while F4 (July) was the least diverse (Table 2). The Cyt b analysis showed that May was the most diverse month followed by June. F2 (May) was the most diverse followed by F3 (June), while F7 (August) was the least diverse (Table 2). Generally, the composition and number of species varied among the sampling seasons and sites. The highest number of species was found in July followed by June. The maximum numbers of species were detected in areas around Lingkun Island and Qidu Island.

### 3.3.4. Comparison of Assessment Tools and Markers

A total of 22 samples of 12S and 18 samples of Cyt b were successful amplified. The total number of sequences read counts passed quality control per library was 784,064 (99.38%) for 12S and 433,420 (88.87%) for Cyt b. After demultiplexing, 757,888 (12S) and 353,581(Cyt b) sequences resulted for taxonomic analysis. A large proportion of Cyt b sequences (61.8%) had no species identity because they had no BLAST hits mostly due to lack of reference sequences available on the GenBank database, while few of 12S (0.01%) had no BLAST hits. The obtained results indicated that 12S marker was more efficient in identifying fish species than Cyt b.

Based on molecular and morphology analysis, a total of 145 species were identified in Oujiang River Estuary, belonging to 57 families and 105 genera. In total, 128 taxa were identified to species level, 11 to genus level and 6 to family level. Based on morphological criteria, 38 taxa were observed from all the samples collected in all these months (55 subsamples) representing 16 families and 27 genera. The 12S metabarcoding dataset identified 82 taxa from 22 subsamples belonging to 46 families and 83 genera. The Cyt b metabarcoding dataset detected 46 taxa from 18 subsamples representing 34 families and 31 genera. The 12S detected 67, Cyt b detected 36, and morphology identified 24 unique species. The number of species in common revealed by the 12S and morphology, by the 12S and Cyt b and by the Cyt b and morphology were eight, four and three, respectively. The three tools identified three species in common (Figure 5 and Supplementary Table S1).

**Table 2.** 12S/Cyt b OTU statistics and alpha diversity indices for ichthyoplankton in Oujiang River Estuary. The value in parenthesis indicates sampling month (4–8 represents April–August). 12S—non-bold values above the slash and Cyt b—bold values below the slash.

| | OTU Statistics | | | OTU Diversity and Abundance | | |
|---|---|---|---|---|---|---|
| Sample | Number of Clean Reads | Number of OTUs | Identified OTUs (%) | Chao1 | Simpson | Shannon |
| F5 (4) | 57,291/**39,856** | 201/**116** | 99.94/**2.40** | 236.04/**136.31** | 0.79/**0.11** | 2.97/**0.61** |
| F3 (5) | 35,468/**37,845** | 126/**137** | 99.99/**83.10** | 198.06/**168.32** | 0.19/**0.46** | 0.79/**1.54** |
| F5 (7) | 19,155 | 165 | 99.94 | 201.12 | 0.83 | 3.32 |
| F5 (8) | 32,023/**10,867** | 124/**98** | 99.95/**5.90** | 148.23/**205.63** | 0.59/**0.25** | 1.94/**1.2** |
| F3 (4) | 3559/**21,629** | 128/**73** | 99.99/**97.7** | 179.75/**143.2** | 0.57/**0.17** | 1.75/**0.8** |
| F3 (6) | 15,119/**11,770** | 69/**119** | 99.99/**35.70** | 128.5/**150.71** | 0.14/**0.54** | 0.58/**1.86** |
| F2 (5) | 49,057/**5850** | 175/**105** | 99.98/**55.00** | 246.32/**150.56** | 0.68/**0.74** | 2.15/**2.85** |
| F2 (6) | 24,758/**8542** | 118/**88** | 99.98/**21.00** | 151.79/**113** | 0.55/**0.41** | 1.74/**1.65** |
| F2 (7) | 92,620/**13,868** | 184/**54** | 99.99/**9.30** | 203.12/**67.2** | 0.66/**0.27** | 2.24/**1.11** |
| F3 (7) | 67,985 | 214 | 99.99 | 254.53 | 0.74 | 2.42 |
| F3 (8) | 21,767/**7532** | 191/**62** | 99.97/**94.60** | 262.36/**83.11** | 0.81/**0.27** | 2.92/**1.19** |
| F4 (5) | 56,326/**9371** | 174/**77** | 100/**97.90** | 244/**100.21** | 0.69/**0.63** | 2.29/**2** |
| F4 (6) | 17,017 | 117 | 100 | 247.71 | 0.42 | 1.67 |
| F4 (7) | 49,670/**10,904** | 96/**31** | 100/**98.00** | 137.35/**41.5** | 0.02/**0.12** | 0.15/**0.57** |
| F4 (8) | 10,946 | 53 | 100 | 80.08 | 0.04 | 0.23 |
| F9 (5) | 31,835 | 135 | 100 | 227.81 | 0.32 | 1.14 |
| F9 (6) | 17,341/**11,816** | 154/**86** | 99.98/**12.90** | 197.56/**123.19** | 0.7/**0.32** | 2.82/**1.25** |
| F9 (7) | 38,062/**90,137** | 137/**213** | 99.99/**34.30** | 168.95/**228.4** | 0.71/**0.56** | 2.32/**1.9** |
| F7 (4) | 14,804/**15,338** | 98/**47** | 100/**18.10** | 152.38/**58.25** | 0.77/**0.34** | 2.41/**1.33** |
| F7 (8) | 11,196/**21,467** | 111/**34** | 99.95/**3.20** | 144/**45** | 0.61/**0.08** | 2.35/**0.39** |
| F7 (6) | 45,304/**14,296** | 131/**61** | 100/**18.20** | 136.83/**70** | 0.68/**0.34** | 2.18/**1.25** |
| F9 (8) | 14,553/**9977** | 167/**94** | 99.60/**9.90** | 238.5/**172** | 0.71/**0.26** | 2.88/**1.2** |
| F5 (5) | **12,316** | **98** | **5.30** | **112.29** | **0.24** | **1.1** |

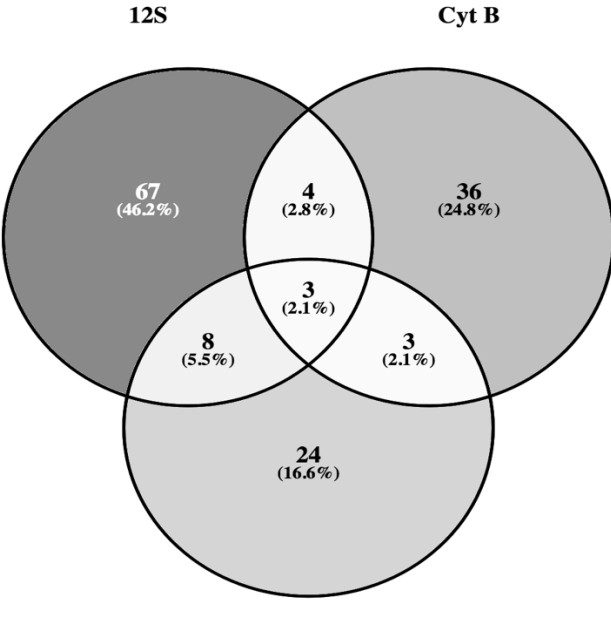

**Figure 5.** Venn diagram showing the overlap numbers of species identified by morphology and molecular tools in Oujiang River Estuary.

## 4. Discussion

### 4.1. Species Identification and Composition

Fish can be identified using distinguishable morphometric and meristic characteristics; however, the latter are typically used for quick identification [27]. Morphological features

commonly used to identify adult fish species are absent at the early development stages, making ichthyoplankton identification more tedious and difficult [28,29]. The species spawn in the area were freshwater, coastal and estuarine related, including Mugilidae, Cyprinidae, Sciaenidae, Lateolabracidae, Gobiidae and pelagic species such as Engraulidae indicated by their high presence.

DNA metabarcoding analysis of ichthyoplankton from Oujiang River Estuary was successful in discerning several fish species and provided biodiversity and abundance data within and between communities. The study identified fish species that commonly and rarely inhabit Oujiang River Estuary. The detected fish species matched previous observations [30–34]. The highest abundance and composition of dominant and common egg and larva species reflected the spawning localities and seasons of adult fish stocks in Oujiang River Estuary. The observed number of unidentified OTUs due to lack of BLAST hit can be explained by sequence data gaps in GenBank database [35]. These findings demonstrated that DNA metabarcoding is a suitable tool for analyzing and monitoring a large scale of pooled samples, due to its ability to produce and detect millions of DNA reads that allow concurrent species identification and analysis [9,36].

### 4.2. Ichthyoplankton Community Structure and Diversity Patterns

The results that most of fish eggs and larvae in Oujiang River Estuary were caught consistently during every sampling month, with the highest catch of eggs and larvae in June and July, indicating that fish species reproduce throughout all five of the months. Fish egg and larva catch increased from April to June and decreased from July, indicating that June is the spawning peak in the river. The findings are consistent with the reports that many fishes in Oujiang River Estuary spawn in June, July and August [31,32]. The community composition, diversity, spatial and temporal distribution of ichthyoplankton varied among sampling sites and months, as revealed by 12S, Cyt b, and morphology. The variation could be the result of changes in oceanographic conditions, specifically a rise in water temperature that favored spawning activities for many fish species [37].

### 4.3. Comparison of Assessment Tools

There was a difference in the PCR amplification success between 12S (<100 bp) and Cyt b (<460 bp) markers. The 12S was more successfully amplified than the Cyt b gene. This could be an effect of the size for the targeted barcode markers [18]. Rees et al. [19] urged that DNA degradation and mismatch of PCR primers in the DNA binding sites affected DNA amplification process that subsequently affected DNA sequencing success. Our results that suggest the 12S marker is more efficient in detecting fish species than Cyt b are in consistence with Hänfling et al. [18]. In this study, 61% of Cyt b and 0.06% of 12S OTUs were unidentified. The variability in species detection could be due to the difference in reference sequences available on GenBank database, and fragment size and persistence. The complete set of fish references obtained for this study included a total of 30,719 (12S) and 4211 (Cyt b) sequences available in GenBank database, thus Cyt b references lacked for many species. Cyt b DNA metabarcoding was therefore unable to detect some common and dominant fish species detected by 12S and morphology in common. A lack of suitable GenBank databases and barcode misidentification accounted for a large proportion of unidentified OTUs. A small proportion of disparity were probably derived from PCR and sequencing errors that could be avoided by improving read preprocessing and quality filtering [12,38,39].

The findings demonstrated that NGS-based metabarcoding is a suitable approach for assessing and analyzing a pooled sample of ichthyoplankton communities. Despite that morphological and molecular species identification often disagree each other but display a certain common degree of taxonomic overlapping [7], each approach can miss the taxa identified by the other [40]. The correct ichthyoplankton identification to species level is possible under molecular identification tools [28]. With suitable primer selection [12], the power of DNA metabarcoding to detect fish species is superior to all conventional fish

assessment and monitoring methods. The results pointed out the potentials and bottlenecks of DNA metabarcoding in identifying fish eggs and larvae and emphasized on the importance of combining molecular and morphological tools in assessing ichthyoplankton community structures.

## 5. Conclusions

The study addressed key issues associated with fisheries management and conservation by providing data regarding fish spawning localities and seasons. Despite a relatively small-scale assessment survey, confidence can be gained in generalizability of spawning seasons is between May and August based on the spatial and temporal analysis. Apparently, DNA metabarcoding is a promising approach for ichthyoplankton ecological and biological survey that expands our current knowledge of fisheries resources. As the 12S genetic marker presented higher species coverage and detection than the Cyt b, the study highlighted the importance of having a complete and accurate reference database for better and more accurate species detection. Generally, the findings reported here provide another key attempt towards the development of powerful and cost-effective ichthyoplankton identification tool, as well as opportunities to overcome the high cost and time consumption in morphological identification.

**Supplementary Materials:** The following supporting information can be downloaded at: https://www.mdpi.com/article/10.3390/d14121111/s1, Table S1. List of identified fish species in Oujiang River Estuary; taxonomy description; identification method; sampling month.

**Author Contributions:** All authors contributed to the study conception and design. Material preparation, data collection and morphological analysis were performed by R.J. DNA extraction, PCR and molecular analysis were performed by J.L.L., Y.C. and R.J. supervised and acquired fundings for the project. The first draft of the manuscript was written by J.L.L. and all the authors commented on previous versions of the manuscript. All authors have read and agreed to the published version of the manuscript.

**Funding:** This study was supported by the National Key R&D Program of China (Grant/Award Numbers: 2018YFD0900903 and 2018YFD0900904) and Zhejiang Ocean University Independent Voyage for Sophisticated Ocean Front and Fisheries Investigation (SOPHI).

**Institutional Review Board Statement:** Not applicable.

**Data Availability Statement:** Cyt b gene: https://www.ncbi.nlm.nih.gov/bioproject/PRJNA911478/; 12S gene: https://www.ncbi.nlm.nih.gov/bioproject/PRJNA911479/.

**Acknowledgments:** We are grateful to Rui Yin and Jing-Xiang Liang of Zhejiang Ocean University for their assistance in sampling and data collection. We thank the anonymous reviewers for their critical and insightful comments on the manuscript.

**Conflicts of Interest:** The authors declare that they have no conflict of interest to declare.

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
