# Peer review of "High-Throughput DNA Metabarcoding as an Approach for Ichthyoplankton Survey in Oujiang River Estuary, China"

_diversity, doi:10.3390/d14121111_

Round 1

Reviewer 1 Report

The manuscript of Jiang et al. depicts the fish diversity in eggs and ichthyoplancton samples collected along 6 stations of the Oujiang River estuary. Authors used an integrative approach to identify their specimens owing to morphological characters and genetic diversity of 2 mitochondrial markers. If this survey is interesting and propose some methodological novelties for first life stage of fishes investigation that deserve to be published my overall feelings is that analyses and results interpretation are too rough.

in the M&M no any reference in 2.2 to any species identification keys are mentioned whereas specimen identification is clearly challenging for eggs and ichthyoplancton as demonstrate by Ko et al. 2013. 

In the paragraph 2.3 it is not clear to me what is name samples, subsamples, ichthyoplankton. Does it mean that authors sort at least 15 individuals from a larvae samples, extract individually the genomic DNA and then pooled all DNA per samples for a metabarcoding analyses? Please clarify.

In 2.3.2 does authors pooled DNA extraction of verious species for the barcoding or use DNA extracted from a batch of samples? Please clarify.

line 138: if I understand authors fixed the sequence divergence at 5% to assign an OTU to a species name. If correct, is there a risk of species admixture/confusion? This could be evaluated by counting the number of cases where more than one species name are associated to GB sequences and where these sequences present less than 5% divergence? Anyway, if DNA barcodes proved their utility for specimen identification authors have to be aware that this approach has some limitation. Maybe they could take a look to the paper of Pham et al. 2022 that used DNA barcoding for fish juvenile specimen ID.

L140 change "OUT" for "OTU"

L145. How authors estimated the beta diversity, did they use number of copy of the different OTU as species abundance or a combination of molecular and morphological identification which seems an issue as the level of resolution is not the same. Please give details.

Results

Paragraph 3.1.1 I am not a specialist but such rate of specimen identification to the species taxonomic rank seems really high

Paragraph 3.2 Presentation of spatial data is very limited, nothing is said about spatial distribution of species

Paragraph 3.3.2 Rate of assignment to a species name seems extremely high using the 12S dataset. Such assignation rate of an OTU to a species was not observed in eDNA survey in similar habitat see Durand et al. 2022 or Abidin et al. 2022. In these study they stressed impact of taxonomical gap in GB library that limit specimen identification accuracy.

Table 3. I am unable to understand how it is possible to "miss" some species using the 12S, species that have been listed using the cytob marker. I checked GB and clearly Terapon jarbua, Coila lindmani, Stolephorus waitei, A. iridescencens etc... have their 12s sequence available in GB. This results seems incoherent and explanation / interpretation in the discussion as sequence gap in GB is not verified.

L313. Interpretation of diversity variation along space and time is really to limited. I understand that the aim of manuscript is to promote the metabarcoding approach for specimen ID but I feel that much more could be done to demonstrate power of this approach. Why no any environmental data is presented ? At least temperature and salinity on station sampled? Presence in the species list o freshwater species like cyprinids and marine species is clearly surprising if no environmental distinction is done among station investigated. See Durand et al. 2022, their is not one community in the Meko,g estuary but at least two depending of the salinity condition. I guess it is the same considering the sampling map but nothing is said about that in the discussion.

P326. Please explain how it is possible to access to the raw dataset in NCBI? Is there a DOI?

At present state I can only encourage authors to pay more attention to their dataset and spend more time to analyse and interpret their data.

Ref.

Abidin et al. 2022 Splash, splash. Who’s there? Advantages and limitations of the environmental DNA (eDNA) metabarcoding in assessing megadiverse but poorly known tropical community of fishes

Durand et al. 2022. Fish Diversity along the Mekong River and Delta Inferred by Environmental-DNA in a Period of Dam Building and Downstream Salinization

Ko et al. 2013 Evaluating the Accuracy of Morphological Identification of Larval Fishes by Applying DNA Barcoding

Pham et al. 2022 Diversity of fishes collected with light traps in the oldest marine protected area in Vietnam revealed by DNA barcoding

Reviewer 2 Report

1 Introduction

══════════════

  This papers combine a metabarcoding and morphological approach to

  characterize ichtyoplankton in the Oujiang River Estuary. For the

  morphological data 11 locations were sampled over 4 months and 6 of

  these locations were also subjected to metabarcoding analysis. In

  general there were limited correlation in species detected by the

  different methods, suggesting that one should be careful making strong

  statements regarding fish species composition based an any single

  method.

2 General comments

══════════════════

  The scope of the study is a large-scale study and ending up with more

  than 30 thousand ichtyoplankton means that one have the potential to

  obtain a comprehensive overview for the studied area. There are two

  main issues with the current manuscript.

  1. Material and methods is not detailed enough to make it feasible for

     redo the analysis and end up with the same results. Since

     reproducibility is central to research this is a major issue. I

     suggest that details regarding parameters used to create reference

     databases, as well as actual commands use to run the actual

     analysis are made available via appendix or link to online lab

     resource. Together with links to raw data deposited in public

     databases this make it possible for the research community to

     validate and make use of the data put forward. As it stands now it

     is not possible to evaluate the results as an external reviewer.

  2. The discrepancy between the two genetic markers needs a more

     comprehensive explanation. The fact in essence all species that are

     found with the 12S data is absent from the cytb data even though

     complete mitochondrial sequence data for these species is available

     at NCBI is puzzling. The same goes for Moringua microchir that is

     only found in the cytb data, representing more than 20% of

     sequences in that data and is not found at all with the 12S marker.

3 Specific remarks

══════════════════

  • Using 3 dimensional pie-charts is not helpful to the reader. It is

    in general hard to estimate proportions in pie-charts and even

    harder if they are in 3d for no apparent reason. I would recommend

    using waffle charts, or if pie-charts are preferred remove any 3d

    attempt.

  • Row 87-90: Was it a random set of 15 individuals selected and those

    were extracted sample by sample or as a pool? If the later is true

    why not use all material for DNA extraction. If it is too much

    material one could crush all material, mix with suitable buffer and

    sub sample that mix for DNA-extraction.

  • Why select such a short piece from 12S. The MyFish primers which are

    more or less standard today is close to 200bp and even this piece

    suffers from being too short to be able to separate closely related

    species or genera.

  • Why cytb instead of CO1 for the longer piece of DNA. CO1 has much

    more data in reference databases and will hence be more suitable for

    this type of analysis

  • A Venn like diagram would be a nice way to illustrate differences

    and similarities between the three data sets. The same goes for

    patterns over time. The data put forward in table 3 would fit in two

    of such figures and the table can be made accessible as a table in a

    supplement.

4 Minor comments, typos and suggestions

═══════════════════════════════════════

  • Row 172: sea bass, anchovy should have scientific names

  • The genus 'Liza' is at least an NCBI referred to as 'Planiliza'

  • Oncorthynchus masou in figure 4 is that a misspelled Oncorhynchus

    masou?

  • O. masou is irrespective of spelling not listed in table 3.

  • Row 256 table text. I do not follow the description here. There is

    no samples from January and December so why are you listing that

    here?

  • Row 294-297 is this really true? If the estimates are completely

    dependent on method of choice it will be difficult to really say

    something about the actual species distribution.

Round 2

Reviewer 1 Report

I read revised manuscript and author's letter and I don't have any specific comments. The manuscript is suitable for Diversity, if editor considers english as correct (I don't feel qualified to judge this criteria).

Author Response

Thank you for your constructive comments on the manuscript.